# Reducing the risk of transfusion-transmitted infectious disease markers in blood and blood component donations: Movement from remunerated to voluntary, non-remunerated donations in Lithuania from 2013 to 2020

Vytenis Kalibatas[1]*, Lina Kalibatienė[2]

1 Department of Health Management, Lithuanian University of Health Sciences, Kaunas, Lithuania,
2 Department of Anaesthesiogy, Lithuanian University of Health Sciences, Kaunas, Lithuania

☯ These authors contributed equally to this work.

* vytenis.kalibatas@lsmuni.lt

**Data Availability Statement:** All relevant data are within the article and its supporting information.

## Abstract

Lithuania has a long history of remunerated donations. The first steps towards voluntary, non-remunerated blood and blood component donations started in 2004. Lithuania achieved 99.98% voluntary non-remunerated donations (VNRDs) in 2020. This study aimed to assess the risk of transfusion-transmitted infectious (TTI) disease markers for remunerated donations in comparison with VNRDs in Lithuania from 2013 to 2020. Data were obtained from the Lithuanian Blood Donor Register. The prevalence was calculated as the rate between the number of confirmed positive results for all TTI disease markers (serological anti-HCV, HBsAg, Ag/anti-HIV 1 and 2, and syphilis, and/or HCV, HBV, and HIV-1 NAT) per 100 donations. The relative risk of infectious disease markers for remunerated donations was then estimated. In total, 796310 donations were made. Altogether, 2743 donations were positive for TTI markers as follows: HCV, 1318; HBV, 768; syphilis, 583; and HIV 1 and 2, 74. The prevalence of confirmed TTI markers were 2.86, 0.97, 0.18, and 0.04 per 100 first-time remunerated donations, first-time VNRDs, repeat remunerated donations, and repeat VNRDs, respectively. Remunerated first-time and repeat donations had a statistically higher prevalence of TTI disease markers than VNRDs. First-time and repeat remunerated donations had statistically significantly higher relative risks of confirmed TTI disease markers than VNRD. In conclusion, the risks of TTI disease markers for remunerated first-time and repeat blood and its component donations are significantly higher than those for VNRDs.

## Introduction

The Melbourne Declaration on 100% Voluntary Non-Remunerated Donation of Blood and Blood Components called for action from all governments to achieve such a goal by 2020 as the cornerstone of their blood policies [1]. The World Health Organization (WHO) revised

**Funding:** The author(s) received no specific funding for this work.

**Competing interests:** The authors have declared that no competing interests exist.

the goal of achieving 100% voluntary, non-remunerated donations (VNRDs) and establishing a sustainable voluntary (non-remunerated and regular) blood donor panel for low-risk populations by 2025 in some areas [2].

Lithuania, a member state of the European Union (EU) since 2004, has sought to change the blood collection system from remunerated to safer voluntary, non-remunerated blood and blood component donations. According to the Lithuanian government's regulation, upon a blood donor's request, the donor was paid €11.6 (40 Lithuanian litas at that time) in cash from the national budget funds until 2014, or €12.0 from year 2015 to 2020 [3]. Such payment is considered compensation for travel supplementary nourishment expenses; however, it does not fall under the definition of VNRDs [4, 5]. The donor can decide to take compensation in cash just before blood or blood component donation up to now.

The first steps towards voluntary, non-remunerated blood and blood component donations started with ordering the adoption of the national program of promoting voluntary blood donorship in 2004 [6], which happened the following year [7]. Since 2006, national programs promoting voluntary blood donorship have been adopted for longer periods: 2006–2015 [8], and 2016–2020 [9]. The national programs include VNRD advertisement companies, public education, and information measures on the advantages of VNRD, small tokens, souvenirs, and refreshments for VNRDs; supply measures for the promotion of VNRDs dedicated to the staff of blood establishments and health care institutions, members of donors' associations, and other non-government organizations (NGOs), which organize and promote VNRDs; and youth education. The expenses of these programs were covered by the Compulsory Health Insurance Fund. Blood establishments were responsible for the planning and administration of the VNRD activities and expenses at the institutional level, and the provision of yearly reports to the Ministry of Health (MoH), based on three main indicators: the number of voluntary, non-remunerated blood and blood component donors, the number of voluntary, non-remunerated blood and blood component donations, and its proportion from the total number of donations. The MoH chaired the program's coordination board, which was responsible for the distribution of funds to blood establishments and NGOs, analysis of the implementation at the institutional and national levels, and supervision on goal attainment. The national program for promoting voluntary blood donorship for 2006–2015 initially set up the following targets: to achieve 20% of VNRDs from the total number of blood and blood component donations at the end of 2007, 35% at the end of 2010, 70% at the end of 2013, and 98% at the end of 2015 (later changed to 38% at the end of 2013, and 55% at the end of 2015). The national program promoting voluntary blood donorship for 2016–2020 targeted a gradual increase in the proportion of VNRDs annually, from 60% at the end of 2016, to 70%, 80%, 90%, eventually reaching 100% at the end of 2020.

Practical and scientific discussions and arguments for and against payment for blood and its component donors are still ongoing. The WHO expert group states that VNRDs are the cornerstone of a safe and sufficient blood supply, and is the first line of defense against the transmission of infectious diseases through transfusion. Informed and regular VNRDs from low-risk populations show a lower risk of HIV and other transfusion-transmissible infections than paid and family/replacement donors [10]. This position is supported by other international authorities (Council of Europe, EBA) [11, 12], although a recent meta-analysis performed by Bruers et al. [13] revealed that the effects of monetary incentives on blood quality remain uncertain, and available observational studies and field experiments show inconclusive evidence that paid donor blood is less safe. Moreover, Farrugia et al. [14] declared that blood-borne infections cannot be automatically and causally linked to the composition of donor populations on the basis of compensation status, while Shearmur [15] concluded no good basis for rejecting whole blood for money.

## Objectives

This study aimed to assess the risk of infectious disease markers for remunerated blood and its component donations, and analyze the prevalence of HIV 1 and 2, HBV, HCV, and syphilis markers per 100 remunerated and voluntary, non-remunerated, and first-time and repeat blood and its component donations and assess the relative risk of TTI disease markers of remunerated first-time and repeat donations in Lithuania from 2013 to 2020.

# Materials and methods

All blood and blood component donations in Lithuania from 2013 to 2020 were analyzed. The data on first-time and repeat remunerated and VNRDs, as well as the number of confirmed positive human immunodeficiency virus (HIV 1 and 2), hepatitis B virus (HBV), hepatitis C virus (HCV), and syphilis cases among those donations were obtained from the annual publications of the blood donors´ register [16–23]. As per local legislations, this survey is not the subject of bioethical regulation, because generalized and fully anonymized, and publicly available data (as opposed to collecting the personal data directly from research participants, medical health records, or archived samples) has been used. Kaunas Regional Biomedical Research Ethics Committee reviewed the methodology of this survey and deemed the investigation an evaluation of service, not requiring review by an ethics committee, as the direct object of this study is not a specific person and/or person´s health. The analysis of de-identified, fully anonymized, generalized, and publicly available data did not constitute human subjects research and did not require participant consent.

First-time donation is defined as a donation from a person who has never previously donated blood or its components. A repeat donation is a donation from a person who repeatedly or routinely donates blood or its components any blood establishment in Lithuania. The donation of blood or blood components, for which the donor receives monetary compensation, is considered a remunerated donation. Donation for which the donor is not monetarily compensated is considered VNRDs.

Blood and blood component donations were tested for serological anti-HCV, HBV surface antigen (HBsAg), Ag/anti-HIV 1 and 2 (ELISA tests) and syphilis (passive haemagglutination assay and chemiluminescent microparticle immunoassay tests). If the result of a serological screening was positive or uncertain, the individual sample test was repeated. If any of the repeated test was positive or uncertain, the following confirmatory tests were used: HBsAg neutralization test for HBV, Anti-HCV immunoblot test for HCV, Western blot and/or HIV 1–p24 antigen and/or HIV 2 ELISA for HIV 1 and 2, and Treponema Pallidum haemagglutination (TPHA) test for syphilis, respectively. Seronegative donations are tested for HCV, HBV, and HIV-1 infections using the nucleic acid test (NAT). If the result of the individual sample was reactive for HIV RNA, HCV RNA, or HBV DNA, the confirmatory quantitative test was performed [24]. Donations with confirmed infectious disease markers (either serological or NAT) were considered as positive.

At the beginning of 2013, there were four blood establishments in Lithuania: a public institution National Blood Center, two university hospital-based blood establishments, and one private, for-profit blood establishment. The latter stopped its activity in April 2013, according to the decision of the State Health Care Accreditation Agency under the Ministry of Health. From that time, the Lithuanian blood system was represented by public blood establishments only.

## Statistical analysis

Prevalence was calculated as the rate between the number of confirmed positive results for infectious disease markers (serological syphilis, anti-HCV, HBsAg, and anti-HIV-1/2, and/or HCV, HBV, HIV-1 NAT) per 100 donations.

The z test was used to compare the two proportions. The relative risk (risk ratio) was estimated to compare the relative risk for infectious disease markers for remunerated and voluntary non-remunerated donations, that is, the probability of occurrence of confirmed infectious disease markers. A *(a+b)/c(c+d)* cross-tabulation was used to estimate the relative risk (risk ratio). Data analysis was conducted using computer software (SPSS, version 26.0). Observations were considered statistically significant if P <0.05.

## Results

Fig 1 shows the annual number of remunerated, voluntary, and non-remunerated blood and blood component donations in Lithuania from 2013 to 2020.

Table 1 presents the annual distribution of blood and blood component donations according to donation type.

In 2013, 0.75% (682 of 90969) donations were confirmed positive for transfusion-transmitted infectious (TTI) disease markers; in 2014–0.53% (493 of 93380); in 2015–0.44% (445 of 100512); in 2016–0.32% (336 of 105175), in 2017–0.30% (303 out of 100008); in 2018–0.19% (199 of 102487), in 2019–0.15% (164 of 107718) and in 2020–0.13% (121 of 96061). The number of donations with confirmed infectious disease markers according to the donation type in Lithuania from 2013 to 2020 is provided S1 Table.

The prevalence of confirmed HCV, HBV, HIV 1 and 2, and syphilis markers per 100 donations from 2013 to 2020, according to donation type, is presented in Table 2.The prevalence of confirmed HCV markers was significantly higher in first-time remunerated donations than in VNRDs (in years 2013–2017) as well as in repeat remunerated donations compared with VNRDs (in years 2013–2018). The prevalence of confirmed HIV 1 and 2 markers was significantly higher in first-time remunerated donations than in VNRDs (in years 2013,2015,2016), while the prevalence of confirmed syphilis markers was significantly higher in first-time remunerated donations than in VNRDs (in years 2013–2017). A statistically significantly higher

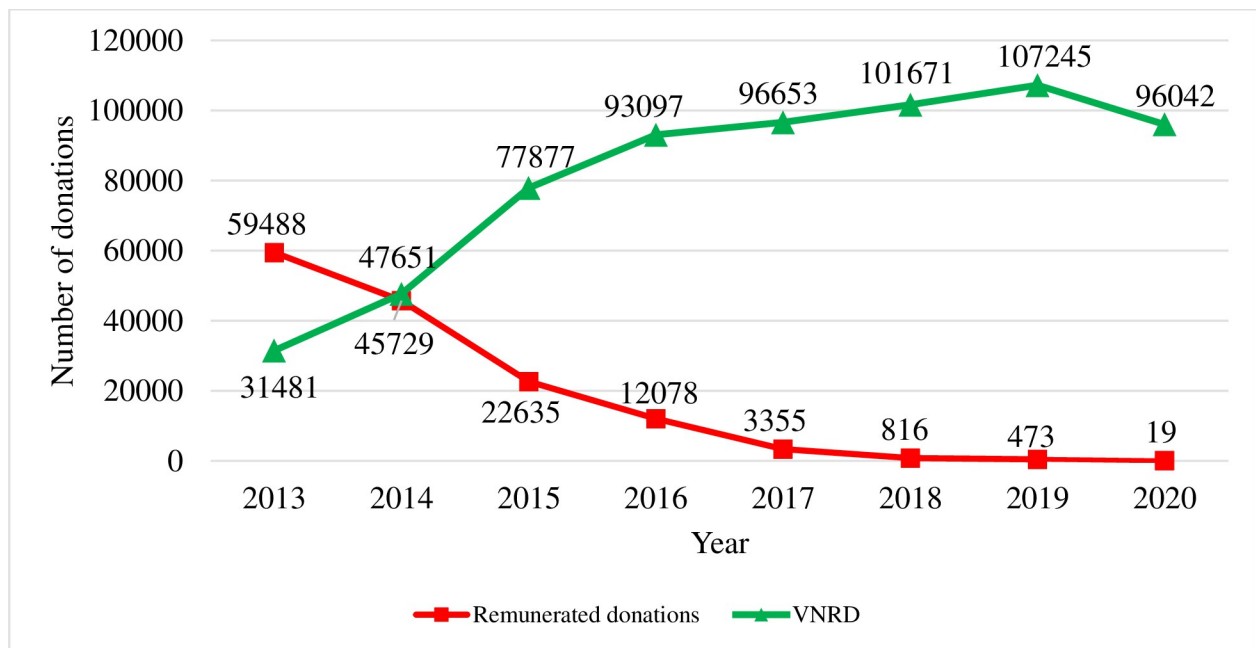

**Fig 1. Remunerated and voluntary, non-remunerated blood and blood component donations in Lithuania from 2013 to 2020.**

**Table 1. The annual distribution of blood and blood component donations according to the donation type from 2013 to 2020.**

| Year | Donations | | | | | | | | | |
|---|---|---|---|---|---|---|---|---|---|---|
| | Remunerated | | | | Voluntary, non-remunerated | | | | All | |
| | First-time | | Repeat | | First-time | | Repeat | | | |
| | N | % | N | % | N | % | N | % | N | % |
| 2013 | 18480 | 20.31 | 41008 | 45.08 | 15143 | 16.65 | 16338 | 17.96 | 90969 | 100.0 |
| 2014 | 6314 | 6.76 | 39415 | 42.21 | 18754 | 20.08 | 28897 | 30.95 | 93380 | 100.0 |
| 2015 | 1940 | 1.93 | 20695 | 20.59 | 28053 | 27.91 | 49824 | 49.57 | 100512 | 100.0 |
| 2016 | 1125 | 1.07 | 10953 | 10.41 | 24618 | 23.41 | 68479 | 65.11 | 105175 | 100.0 |
| 2017 | 227 | 0.23 | 3128 | 3.13 | 21874 | 21.87 | 74779 | 74.77 | 100008 | 100.0 |
| 2018 | 1 | 0.001 | 815 | 0.795 | 18970 | 18.509 | 82701 | 80.695 | 102487 | 100.0 |
| 2019 | 2 | 0.002 | 471 | 0.437 | 17509 | 16.254 | 89736 | 83.307 | 107718 | 100.0 |
| 2020 | 0 | 0.0 | 19 | 0.02 | 10812 | 11.26 | 85230 | 88.72 | 96061 | 100.0 |

prevalence of HBV infection in first-time remunerated donations than in first-time VNRDs was identified in 2017. There was only one first-time remunerated donation in 2018, two in 2019, and one in 2020, and 19 repeat remunerated donations in 2020, thus due to insufficient sample sizes comparisons with VNRDs are not applicable.

The prevalence of all confirmed infectious disease markers per 100 donations from 2013 to 2020, according to donation type, is presented in Table 3. As shown in the Table 3, the prevalence of confirmed infectious disease markers in first-time remunerated donations was significantly higher compared with first-time VNRD from 2013 to 2017, when comparing each year. The prevalence of infectious disease markers in repeat remunerated donations was significantly higher than that in repeat VNRD in all years, except in 2018.

Pooled data from 2013 to 2020 showed that there were 796310 donations. Of these, in 2743 donations there were confirmed TTI markers: 1318—HCV, 768–HBV, 583–syphilis, and 74 – HIV 1 and 2. A total of 804 confirmed TTI cases (HCV—490, HBV– 123, HIV 1 and 2–17, and syphilis -174) were found in 28089 first-time remunerated donations. A total of 1509 confirmed TTI cases (HCV, 593; HBV, 583; HIV 1 and 2, 19; syphilis, 314) were found in 155733 first-time VNRD cases. A total of 214 confirmed TTI cases (HCV—154, HBV– 13, HIV 1 and 2–10, and syphilis -37) were found in 116504 repeat remunerated donations. A total of 216 confirmed TTI cases (HCV, 81; HBV, 49; HIV 1 and 2, 28; and syphilis, -58) were found among 495984 repeat VNRD cases. The prevalence of all confirmed TTI markers in first-time remunerated donations was significantly higher than that in VNRDs (2.86 vs 0.97 per 100 donations; z = 26.20; P< 0.0001). The prevalence of all confirmed TTI markers in repeat remunerated donations was significantly higher than that in VNRDs (0.099 vs. 0.044 per 100 donations; z = 16.25; P< 0.0001).

The probability for the confirmed TTI markers was estimated to be statistically significantly higher in first-time remunerated donations than in VNRDs in all years, except for 2020. No first-time remunerated donations were made in 2020. Table 4 summarizes the relative risk for confirmed infectious disease markers for first-time remunerated and voluntary non-remunerated donations in Lithuania from 2013 to 2020. The pooled data from 2013 to 2020 showed that the relative risk of confirmed TTI markers in first-time remunerated donations is 2.86 times higher than that in first-time VNRDs (95% CI, 2.6643– 3.1558; z = 54.85; P< 0.0001).

The probability for the confirmed TTI markers was estimated to be statistically significantly higher in repeat remunerated donations than in VNRDs in all years except 2018. Table 5 shows the relative risk of confirmed infectious disease markers for repeat remunerated and

**Table 2. Prevalence of confirmed HCV, HBV, HIV 1 and 2 and syphilis markers per 100 donations according to the donation type from 2013 to 2020.**

| Year/ TTI marker | Prevalence per 100 donations | | | | | | | |
|---|---|---|---|---|---|---|---|---|
| | First-time | | | | Repeat | | | |
| | Remunerated | VNRD | z-score | *P*-value | Remunerated | VNRD | z-score | *P*-value |
| **2013** | | | | | | | | |
| HCV | 1.418 | 0.370 | 9.88 | **<0.001** | 0.159 | 0.031 | 3.96 | **<0.001** |
| HBV | 0.411 | 0.284 | 1.96 | >0.05 | 0.015 | 0.024 | 0.81 | >0.05 |
| HIV 1 and 2 | 0.054 | 0.013 | 1.98 | **<0.05** | 0.012 | 0.00 | 1.41 | >0.05 |
| Syphilis | 0.503 | 0.225 | 4.14 | **<0.001** | 0.039 | 0.031 | 0.48 | >0.05 |
| **2014** | | | | | | | | |
| HCV | 1.996 | 0.528 | 10.69 | **<0.001** | 0.094 | 0.031 | 3.12 | **<0.05** |
| HBV | 0.475 | 0.352 | 1.37 | >0.05 | 0.013 | 0.010 | 0.17 | >0.05 |
| HIV 1 and 2 | 0.048 | 0.011 | 1.79 | >0.05 | 0.008 | 0.007 | 0.10 | >0.05 |
| Syphilis | 0.824 | 0.203 | 7.13 | **<0.001** | 0.030 | 0.021 | 0.77 | >0.05 |
| **2015** | | | | | | | | |
| HCV | 2.887 | 0.385 | 14.45 | **<0.001** | 0.116 | 0.020 | 5.28 | **<0.001** |
| HBV | 0.412 | 0.406 | 0.04 | >0.05 | 0.00 | 0.016 | 1.82 | >0.05 |
| HIV 1 and 2 | 0.103 | 0.021 | 2.13 | **<0.05** | 0.005 | 0.006 | 0.19 | >0.05 |
| Syphilis | 0.876 | 0.257 | 4.85 | **<0.001** | 0.039 | 0.016 | 1.81 | >0.05 |
| **2016** | | | | | | | | |
| HCV | 3.289 | 0.321 | 14.53 | **<0.001** | 0.192 | 0.019 | 8.11 | **<0.001** |
| HBV | 0.533 | 0.431 | 0.51 | >0.05 | 0.018 | 0.015 | 0.29 | >0.05 |
| HIV 1 and 2 | 0.178 | 0.012 | 3.89 | **<0.001** | 0.009 | 0.004 | 0.65 | >0.05 |
| Syphilis | 0.444 | 0.179 | 1.99 | **<0.05** | 0.009 | 0.004 | 0.65 | >0.05 |
| **2017** | | | | | | | | |
| HCV | 3.965 | 0.494 | 7.17 | **<0.001** | 0.160 | 0.023 | 4.74 | **<0.001** |
| HBV | 1.322 | 0.398 | 2.17 | **<0.05** | 0.00 | 0.012 | 0.61 | >0.05 |
| HIV 1 and 2 | 0.00 | 0.005 | 0.10 | >0.05 | 0.00 | 0.008 | 0.50 | >0.05 |
| Syphilis | 3.082 | 0.201 | 9.00 | **<0.001** | 0.00 | 0.009 | 0.54 | >0.05 |
| **2018** | | | | | | | | |
| HCV | 0.00 | 0.300 | - | - | 0.123 | 0.017 | 3.32 | **<0.05** |
| HBV | 0.00 | 0.343 | - | - | 0.00 | | 0.24 | >0.05 |
| HIV 1 and 2 | 0.00 | 0.011 | - | - | 0.00 | | 0.24 | >0.05 |
| Syphilis | 0.00 | 0.169 | - | - | 0.00 | | 0.40 | >0.05 |
| **2019** | | | | | | | | |
| HCV | 0.00 | 0.274 | - | - | 0.212 | 0.008 | 4.70 | **<0.001** |
| HBV | 0.00 | 0.326 | - | - | 0.00 | 0.004 | 0.14 | >0.05 |
| HIV 1 and 2 | 0.00 | 0.017 | - | - | 0.00 | 0.004 | 0.14 | >0.05 |
| Syphilis | 0.00 | 0.177 | - | - | 0.00 | 0.010 | 0.22 | >0.05 |
| **2020** | | | | | | | | |
| HCV | - | 0.351 | - | - | 0.00 | 0.007 | - | - |
| HBV | - | 0.416 | - | - | 0.00 | 0.006 | - | - |
| HIV 1 and 2 | - | 0.00 | - | - | 0.00 | 0.005 | - | - |
| Syphilis | - | 0.176 | - | - | 0.00 | 0.005 | - | - |

voluntary non-remunerated donations in Lithuania from 2013 to 2020. The pooled data from 2013 to 2020 showed that the relative risk of confirmed TTI markers in repeat remunerated donations was 4.21 times higher than that in repeat VNRDs (95% CI, 3.4868–5.0878; z = 14.92; p < 0.0001).

Table 3. Prevalence of confirmed all transfusion-transmitted infectious disease markers per 100 donations from 2013 to 2020 according to donation type.

| Year | Prevalence per 100 donations | | | | | | | |
|------|---|---|---|---|---|---|---|---|
| | First-time | | | | Repeat | | | |
| | Remunerated | VNRD | z-score | P-value | Remunerated | VNRD | z-score | P-value |
| 2013 | 2.39 | 0.89 | 10.51 | < **0.001** | 0.22 | 0.09 | 3.49 | < **0.001** |
| 2014 | 3.34 | 1.09 | 12.10 | < **0.001** | 0.14 | 0.07 | 2.90 | <**0.05** |
| 2015 | 4.28 | 1.07 | 12.17 | < **0.001** | 0.16 | 0.06 | 4.13 | < **0.001** |
| 2016 | 4.44 | 0.94 | 11.04 | < **0.001** | 0.23 | 0.04 | 6.93 | < **0.001** |
| 2017 | 8.37 | 1.10 | 10.13 | < **0.001** | 0.16 | 0.05 | 2.48 | <**0.05** |
| 2018 | 0 | 0.82 | - | - | 0.12 | 0.05 | 0.90 | >0.05 |
| 2019 | 0 | 0.79 | - | - | 0.21 | 0.03 | 2.41 | <**0.05** |
| 2020 | - | 0.94 | - | - | 0 | 0.02 | - | - |

## Discussion

It is noteworthy that the blood collection system in Lithuania provides a unique possibility to compare the prevalence of confirmed infectious disease markers and risks among remunerated and non-remunerated donations. First, donations were performed at blood establishments and covered all blood and blood component donations, except for plasma donations for pharmaceutical or commercial purposes, which were not collected in Lithuania. Second, the payment for donations was organized and covered by the government of Lithuania, providing a unified payment that remained nearly steady (€11.6 in 2013 and 2014, and €12 from 2015 to 2020). Third, the requirements for serological tests and NAT of each donation as well as confirmation of infectious disease markers were set up in the order of the MoH, and thus were the same in all blood establishments for all donations (whole blood, red blood cells, platelets). Fourth, each blood establishment was obliged to provide information on blood and its component donations, donors, testing of infectious disease markers, etc., on a quarterly basis to the blood donor register, which is run by the governmental Institute of Hygiene. Finally, the breaking point of the shift from remunerated to VNRDs was observed from 2013 to 2020, allowed to summarize the findings in Lithuania. This situation provided the unique possibility of comparing the risks of transfusion-transmitted infectious disease markers among blood and blood component donations from 2013 to 2020.

Analyzing the prevalence of confirmed infectious disease markers per 100 donations in Lithuania showed that the highest prevalence of confirmed TTI markers was among first-time

Table 4. Relative risks for confirmed infectious disease markers for first-time remunerated and VNRDs in Lithuania from 2013 to 2020.

| Year | Relative risk* | 95% CI | z value | P-value |
|------|----------------|--------|---------|---------|
| 2013 | 2.68 | 2.21–3.24 | 10.07 | < **0.001** |
| 2014 | 3.06 | 2.53–3.69 | 11.52 | < **0.001** |
| 2015 | 4.00 | 3.15–5.08 | 11.38 | < **0.001** |
| 2016 | 4.72 | 3.49–6.36 | 10.14 | < **0.001** |
| 2017 | 7.63 | 4.87–11.95 | 8.88 | < **0.001** |
| 2018 | 30.31 | 2.73–335.92 | 2.78 | <**0.05** |
| 2019 | 16.14 | 1.28–203.79 | 2.15 | <**0.05** |
| 2020 | - | - | - | - |

*Relative risk value, estimating the risk of confirmed infectious disease marker between first-time remunerated and VNRDs per year

**Table 5. Relative risks for confirmed infectious disease markers for repeat remunerated and VNRDs in Lithuania from 2013 to 2020.**

| Year | Relative risk* | 95% CI | z value | P-value |
|------|---------------|--------|---------|---------|
| 2013 | 2.62 | 1.49–4.59 | 3.36 | <**0.001** |
| 2014 | 2.09 | 1.26–3.48 | 2.84 | <**0.05** |
| 2015 | 2.74 | 1.66–4.51 | 3.96 | <**0.001** |
| 2016 | 5.39 | 3.16–9.20 | 6.18 | <**0.001** |
| 2017 | 3.06 | 1.21–7.77 | 2.36 | <**0.05** |
| 2018 | 2.42 | 0.33–17.53 | 0.87 | >0.05 |
| 2019 | 7.94 | 1.08–58.56 | 2.03 | <**0.05** |
| 2020 | 109.27 | 6.83–1748.59 | 3.32 | <**0.001** |

*Relative risk value, estimating the risk of confirmed infectious disease markers between repeat remunerated and VNRDs per year

remunerated donations. The yearly trend of confirmed TTI markers per 100 donations increased from 2013 to 2017 for HCV, HBV, and syphilis; HIV 1 and 2 increased from 2013 to 2016. It should be noted that the number of first-time remunerated donations decreased during that period (from 18480 donations in 2013 to 227 donations in 2017, with the latter dramatic shrinkage until any donation in 2020), which could explain the increase in prevalence of TTI markers. There was one first-time remunerated donation in 2018, two in 2019, and none in 2020, which could explain the zero prevalence of all TTI markers in first-time remunerated donations during the period 2018–2020.

During the study period, the number of first-time VNRDs fluctuated–from 15143 donations in 2013 to 28053 donations in 2015, followed by a yearly decrease until 10812 donations in 2020. The trend of confirmed TTI markers in first-time VNRDs between the period 2013–2020 remains stable for HCV, in contrast to the decline in HCV infection in first-time donors in high-income OECD countries, including England and Wales, Australia, the Netherlands, the US, and Canada [25]. The prevalence of HBV in first-time VNRDs also does not show a remarkable declining trend, while in Italy [26], the US Red Cross [27], and China [28], there was a decrease in the trends for HBV infections. The prevalence of HIV 1 and 2 infection remained stable, whereas the prevalence of syphilis infection per 100 first-time VNRDs decreased slightly during the study period. Overall, the prevalence of TTI markers in first-time VNRDs was significantly lower than that in first-time remunerated donations in years 2013–2017. The probability for the confirmed TTI markers was estimated to be statistically significantly higher in first-time remunerated donations than in VNRDs in all years, except for 2020. Overall, the pooled data showed that the relative risk of confirmed TTI markers in first-time remunerated donations is statistically significantly 2.86 times higher than that in first-time VNRDs.

There isn't any published data with evidence that repeat VNRDs are safer than repeat remunerated donations in whole Lithuania. A previous study at the largest blood establishment in Lithuania compared the prevalence of TTI in remunerated and VNRDs in two years (2005–2006) period, and estimated a statistically significant higher prevalence of TTI only in first-time remunerated donations than in first-time VNRDs [29]. A later study at the same blood establishment covered 6 years (2005–2010) period, and showed that remunerated first-time and repeat whole blood and platelet seronegative donations have a statistically significant higher prevalence of NAT-positive HBV and HCV markers than voluntary, non-remunerated first-time and repeat donations [30]. Our study showed that the prevalence rate of HCV in

repeat remunerated donations is statistically significantly higher than that in repeat VNRDs each year, except for 2020. It should be mentioned that in 2020, there were only 19 remunerated donations. Overall, the prevalence of TTI markers in repeat VNRDs was significantly lower than that in first-time remunerated donations in years 2013–2017, and in year 2019. The probability for the confirmed TTI markers was estimated to be statistically significantly higher in repeat remunerated donations than in VNRDs in all years except 2018. The pooled data showed that the relative risk of confirmed TTI markers in repeat remunerated donations is statistically significant 4.21 times higher than in repeat VNRDs. Analysis of the trend of confirmed TTI in repeat remunerated donations showed a stable prevalence rate of confirmed HCV markers from 2013 to 2019, while the prevalence of other HBV and HIV 1 and 2 markers was stable, with a slight decrease in syphilis markers in years 2013–2016. In contrast, the prevalence of confirmed TTI markers in repeat voluntary non-remunerated donors decreased during the study period, which is a trend in many high-income OECD countries [25]. The number of repeat remunerated donations has decreased dramatically during the study period, from 41008 donations in 2013 until 19 donations in 2020, whereas the number of repeat VNRDs has increased several times, from 16338 donations in 2013 until 89736 donations in 2019, with a decrease in 2020.

TTI diseases are more frequently detected among newly registered donors and first-time donors than among repeat donors [31]. This statement is fully supported by the results of the present study. Both first-time and repeat remunerated donations are less safe than voluntary non-remunerated donations, and this is not an empirical claim [32]. The experience in Lithuania could indicate that voluntary unpaid donors have the lowest rates of transfusion-associated infections and are the ideal population from which to recruit donors [33]. Finally, the successful step-by-step shift from remunerated to VNRDs (99.98% in 2020) can be explained by the following aspects. First, there is the continuous and active position of the Ministry of Health of Lithuania regarding the goal of achieving 100% VNRD (long-term national programs to promote voluntary blood donorship and control over the achievement of the yearly goals). Second, the decreased value of money received for the donation–the payment remained nearly the same from 2004 to 2020 (€11.6 from 2004 to 2014, and €12 from 2015 to 2020), while average gross earnings and net earnings were €379.40 and €295.71 in 2004; €614.28 and €478.77 in 2010; €7569 and €5848 in 2015; €15242 and €9673 in 2020, respectively. Third, the activity of private, for-profit blood establishments stopped in 2013. The private, for-profit entity made constant pressure to sell more blood components, and thus made payments for the donors, as stated by Petrini [34]. Moreover, the endless fighting for the blood components market share with public, non-profit blood establishments, with the involvement of some governmental institutions and politicians, did not pose confidence with the concept of VNRDs. Finally, the rejection of the private, for-profit blood establishment from the national blood system was in line with the principle of the WHO Expert Group regarding the prevention of the commercialization of donation of blood, plasma, and cellular components and exploitation of blood donors [10], and possibly increased trust of the voluntary, non-remunerated donors to the national blood system.

The blood establishments are obliged to provide information to the donor about positive screening and NAT test results, and the donor shall be prohibited from donating blood if the confirmatory test is positive [35]. There are no requirements to assess the risk behavior of the blood donor in a case of prohibited donation. It probably could better explain why remunerated donors have a higher prevalence and relative risk of confirmed transfusion-transmitted infectious disease markers than VNRDs. During this study we emphasized the fact of remuneration, not the possible reasons of the confirmed TTI.

The limitations of this study are as follows. The first concerns the issue of prevalence as well as risk analyses, which are related to donations, not donors, and possible co-infections among blood and its components donors have not been analyzed The second aspect concerns that the risk of remunerated donations was based on confirmed TTI markers, not post-transfusion infections among recipients of blood components. Third, plasma donations for pharmaceutical or commercial purposes are not performed in Lithuania. Thus, the results of the study are related only to blood and blood component' donations, excluding plasma donations for pharmaceutical purposes.

## Conclusion

Remunerated first-time and repeat blood and blood component donations have a statistically significant higher prevalence of transfusion-transmitted infectious disease markers than VNRDs. There were statistically significant higher relative risks of confirmed transfusion-transmitted infectious disease markers in first-time and repeat remunerated donations than in VNRDs.

## Supporting information

**S1 Table. The number of donations with confirmed infectious disease markers according to the donation type in Lithuania from 2013 to 2020.**
(DOCX)

## Author Contributions

**Conceptualization:** Vytenis Kalibatas, Lina Kalibatienė.

**Formal analysis:** Vytenis Kalibatas, Lina Kalibatienė.

**Investigation:** Vytenis Kalibatas, Lina Kalibatienė.

**Methodology:** Vytenis Kalibatas, Lina Kalibatienė.

**Resources:** Vytenis Kalibatas.

**Supervision:** Vytenis Kalibatas.

**Validation:** Vytenis Kalibatas, Lina Kalibatienė.

**Visualization:** Lina Kalibatienė.

**Writing – original draft:** Vytenis Kalibatas, Lina Kalibatienė.

**Writing – review & editing:** Vytenis Kalibatas.

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
