## [Editor Report · Decision Letter 0]

13 Apr 2022

PONE-D-22-03454Reducing the risk of transfusion-transmitted infectious disease markers in blood and blood component donations: movement from remunerated to voluntary, non-remunerated donations in Lithuania from 2013 to 2020PLOS ONE

Dear Dr. Kalibatas,

Thank you for submitting your manuscript to PLOS ONE. After careful consideration, we feel that it has merit but does not fully meet PLOS ONE’s publication criteria as it currently stands. Therefore, we invite you to submit a revised version of the manuscript that addresses the points raised during the review process.

We look forward to receiving your revised manuscript.

Kind regards,

Amita Radhakrishnan Nair

Academic Editor

PLOS ONE

Journal Requirements:

Additional Editor Comments:

Dear Author,

The work you have done is very good.

But I have certain reservations.

Serological test always carries an element of false positivity. Was any other test done to confirm the positive status of donor?

All repetitions may be deleted.

The results sections has to be re-written and concise

1. Line 31, Please rewrite as HIV 1 and 2.

2. Line 207-215, If for statistical analysis, sample size is insufficient, need not look at the statistics.

3. Year wise statistics has not much relevance. It can be combined together

4. Line 205-208, where sufficient numbers not available, tests of significance carries no value. Such results may be omitted

5. Line 232- not complete

6. Were the TTI reactive donors contact and post test counselling done to assess the risk behaviors?

7. Line 297-98, what could be the reason for this finding?

8. Help of a statistician may be sought for better presentation of the results.

---

## [Author Response · Author response to Decision Letter 0]

9 May 2022

Dear Editor, 

We thank you and the reviewers for a thorough reading and constructive criticism of our manuscript and for the opportunity to revise and resubmit. We are pleased to submit the improved research article “Reducing the risk of transfusion-transmitted infectious disease markers in blood and blood component donations: movement from remunerated to voluntary, non-remunerated donations in Lithuania from 2013 to 2020” for your consideration for publication in PLOS ONE. On the following pages, you will find our response to the Editor comments. 

On behalf of the authors, I thank you for your consideration of this resubmission. We appreciate your time and look forward to your response. 

Sincerely,

Ass. Prof. Vytenis Kalibatas, MD, MPH, PhD (corresponding author)

Department of Health Management,

Lithuanian University of Health Sciences

Tilžės str. 18, LT-47181, Kaunas, Lithuania

E-mail: vytenis.kalibatas@lsmuni.lt

Journal Requirements:

We thank you for the advices on the revision. We followed the templates and changed the figure file according to the requirements. 

2. In your Data Availability statement, you have not specified where the minimal data set underlying the results described in your manuscript can be found. PLOS defines a study's minimal data set as the underlying data used to reach the conclusions drawn in the manuscript and any additional data required to replicate the reported study findings in their entirety. All PLOS journals require that the minimal data set be made fully available. For more information about our data policy, please see http://journals.plos.org/plosone/s/data-availability. Upon re-submitting your revised manuscript, please upload your study’s minimal underlying data set as either Supporting Information files or to a stable, public repository and include the relevant URLs, DOIs, or accession numbers within your revised cover letter. For a list of acceptable repositories, please see http://journals.plos.org/plosone/s/data-availability#loc-recommended-repositories. Any potentially identifying patient information must be fully anonymized. 

We have uploaded our supplementary information in S1 Table. All relevant data are within the article and its supporting information. 

Additional Editor Comments:

1. Serological test always carries an element of false positivity. Was any other test done to confirm the positive status of donor?

 In the manuscript, we have used the word "confirmed", which we treated as "a priori" explanation that we provide the data only on confirmed TTI markers, not initially positive. 

 Anyway, to make the text more understandable for all readers, we added some explanations regarding the confirmatory testing in the "Material and methods" section (125-134 

 lines in Manuscript; 125-133 lines in the Revised Manuscript with Track Changes).

2. All repetitions may be deleted.

 We have removed all repetitions in the "Results" section and included the additional table in the Supporting Information.

3. The results sections has to be re-written and concise.

 We rewrote the "Results" section and concise the text.

4. Line 31, Please rewrite as HIV 1 and 2.

 We corrected the marker in the “Abstract” section. 

5. Line 207-215, If for statistical analysis, sample size is insufficient, need not look at the statistics.

 We deleted sentences in lines 207-215, and added that due to insufficient sample sizes comparisons with VNRDs are not applicable (lines 179-180 in Manuscript; lines 208-209 in 

 the Revised Manuscript with Track Changes).

6. Year wise statistics has not much relevance. It can be combined together.

 Thank you for this comment. We could agree with it if we emphasize a sufficient sample size of remunerated donations in some years. By combining some years (for example, 

 2013-2014; 2015- 2016, etc.), we miss some important aspects of showing the breaking points from prevailing remunerated to VNRDs, and the shift towards the safest donation 

 group - repeat VNRDs, which prevails starting from 2016. On the other hand, combining two or more years does not solve the problem of a sufficient sample size for first-time 

 remunerated donations. In a manuscript, we provide the pooled data, combining all years, but we also think that data presented yearly give some useful information about 

 donations' changes.

7. Line 205-208, where sufficient numbers not available, tests of significance carries no value. Such results may be omitted.

 We deleted sentences in lines 205-205, and added that due to insufficient sample sizes comparisons with VNRDs are not applicable (lines 179-180 in Manuscript; lines 208-209 in 

 the Revised Manuscript with Track Changes).

8. Line 232- not complete.

 We added the year 2020 (line 209 in Manuscript; line 233 in the Revised Manuscript with Track Changes).

9. Were the TTI reactive donors contact and post test counselling done to assess the risk behaviors?

 In answering the question, we provided the additional information in the “Discussion” section (lines 319-325 in Manuscript; lines 394 - 400 in the Revised Manuscript with Track 

 Changes).

10. Line 297-98, what could be the reason for this finding?

 Thank you for pointing it out. The sentence was incorrect: “repeat VNRDs are safer than repeat non-remunerated donations”. We corrected the sentence (lines 272-273 in 

 Manuscript; lines 346 - 347 in the Revised Manuscript with Track Changes).

11. Help of a statistician may be sought for better presentation of the results.

 We consulted with a statistician and made some changes in calculations (z-test instead of chi-square test for comparisons of prevalence), as well as in the interpretation of the 

 results.

---

## [Decision Letter · Decision Letter 1]

2 Nov 2022

Reducing the risk of transfusion-transmitted infectious disease markers in blood and blood component donations: movement from remunerated to voluntary, non-remunerated donations in Lithuania from 2013 to 2020

PONE-D-22-03454R1

Dear Dr.Kalibatas ,

We’re pleased to inform you that your manuscript has been judged scientifically suitable for publication and will be formally accepted for publication once it meets all outstanding technical requirements.

Kind regards,

Mohamed A Yassin, MD

Academic Editor

PLOS ONE

Additional Editor Comments (optional):

Reviewers' comments:

Reviewer's Responses to Questions

**Comments to the Author**

1. If the authors have adequately addressed your comments raised in a previous round of review and you feel that this manuscript is now acceptable for publication, you may indicate that here to bypass the “Comments to the Author” section, enter your conflict of interest statement in the “Confidential to Editor” section, and submit your "Accept" recommendation.

Reviewer #1: (No Response)

2. Is the manuscript technically sound, and do the data support the conclusions?

Reviewer #1: Yes

3. Has the statistical analysis been performed appropriately and rigorously? 

Reviewer #1: Yes

4. Have the authors made all data underlying the findings in their manuscript fully available?

Reviewer #1: Yes

5. Is the manuscript presented in an intelligible fashion and written in standard English?

Reviewer #1: Yes

6. Review Comments to the Author

Reviewer #1: Dear authors in this manuscript you described about the role of voluntary non-renumerated donation in reducing the risk of transfusion transmitted infectious diseases. You responded previous reviewer’s comments very nicely, but there are minor changes can improve the manuscript.

Specific comments:

1. In this manuscript you used the term for HIV used in various different words such as somewhere you used “HIV” only while in soe places you used “HIV 1 and 2”, it would be great if you use the same term “HIV 1 and 2” in whole manuscript.

2. In introduction section line 53-60, the donor was paid for donations (12 euro from year 2015), is this practice is still in place.

3. In material and method line 125, you mentioned that blood donation was tested for serological markers for various TTIs. It would be great if you could mention the method for serological testing such as rapid test or ELISA or Chemiluminescence test.

4. You mentioned Fig 1 and Table 1 I material and method section, I feel that it should be included under result section because these are your findings of the study about annual blood donation.

5. In this manuscript you did not mention about co-infections, kindly mention if there were any co-infections (like donor was reactive for two or more TTIs).

6. In discussion line 237 you mentioned the payment 11.6 euro per donation from 2013 to 2020, while in introduction section line 53-60 you mentioned that payment increased to 12 euro from year 2015. These both are quite confusing statements. Kindly clarify.

7. PLOS authors have the option to publish the peer review history of their article (what does this mean?). If published, this will include your full peer review and any attached files.

Reviewer #1: No

---

## [Editor Report · Acceptance letter]

4 Nov 2022

PONE-D-22-03454R1 

Reducing the risk of transfusion-transmitted infectious disease markers in blood and blood component donations: movement from remunerated to voluntary, non-remunerated donations in Lithuania from 2013 to 2020 

Dear Dr. Kalibatas:

I'm pleased to inform you that your manuscript has been deemed suitable for publication in PLOS ONE. Congratulations! Your manuscript is now with our production department. 

Kind regards, 

on behalf of

Dr. Mohamed A Yassin 

Academic Editor

PLOS ONE